# Mutations in Complex I of the Mitochondrial Electron-Transport Chain Sensitize the Fruit Fly (*Drosophila melanogaster*) to Ether and Non-Ether Volatile Anesthetics

**DOI:** 10.3390/ijms24031843

**Published:** 2023-01-17

**Authors:** Luke A. Borchardt, Amanda R. Scharenbrock, Zachariah P. G. Olufs, David A. Wassarman, Misha Perouansky

**Affiliations:** 1Department of Anesthesiology, University of Wisconsin-Madison, Madison, WI 53792, USA; 2Department of Medical Genetics, University of Wisconsin-Madison, Madison, WI 53706, USA; 3Laboratory of Genetics, University of Wisconsin-Madison, Madison, WI 53706, USA

**Keywords:** mitochondria, electron transport chain, volatile anesthetics, mutations, Complex I, toxicity, isoflurane, halothane, anoxia

## Abstract

The mitochondrial electron transport chain (mETC) contains molecular targets of volatile general anesthetics (VGAs), which places carriers of mutations at risk for anesthetic complications. The *ND-23^60114^* and mt:*ND2^del1^* lines of fruit flies (*Drosophila melanogaster*) that carry mutations in core subunits of Complex I of the mETC replicate numerous characteristics of Leigh syndrome (LS) caused by orthologous mutations in mammals and serve as models of LS. *ND-23^60114^* flies are behaviorally hypersensitive to volatile anesthetic ethers and develop an age- and oxygen-dependent anesthetic-induced neurotoxicity (AiN) phenotype after exposure to isoflurane but not to the related anesthetic sevoflurane. The goal of this paper was to investigate whether the alkane volatile anesthetic halothane and other mutations in Complex I and in Complexes II–V of the mETC cause AiN. We found that (i) *ND-23^60114^* and mt:*ND2^del1^* were susceptible to toxicity from halothane; (ii) in wild-type flies, halothane was toxic under anoxic conditions; (iii) alleles of accessory subunits of Complex I predisposed to AiN; and (iv) mutations in Complexes II–V did not result in an AiN phenotype. We conclude that AiN is neither limited to ether anesthetics nor exclusive to mutations in core subunits of Complex I.

## 1. Introduction

Anesthetic-induced neurotoxicity (AiN) is a potential concern anytime volatile general anesthetics (VGAs) are administered, but certain conditions increase the risk of AiN. Risk factors include the extremes of age, the fragile and/or diseased brain and genetic factors [1]. Among the most relevant risk factors for ‘collateral’ effects of VGAs are interactions with mitochondrial function. The mitochondrial electron transport chain (mETC) contains molecular targets of VGAs and mETC function is dose-dependently suppressed by VGAs at clinically relevant concentrations in ‘wild-type’ mitochondria [2]. Mutations in Complex I of the mETC increase the sensitivity of oxidative phosphorylation to depression by VGAs resulting in anesthetic hypersensitivity [3] and possibly in increased risk for AiN [4,5]. Over 80 mutations mostly affecting Complex I have been identified as causing Leigh Syndrome (LS), a rare incurable neurodegenerative disease resulting in severe disability and early death and associated with increased perioperative morbidity and sensitivity to VGAs [6].

Although LS is a risk factor for perioperative complications, LS patients are frequently exposed to anesthesia for diagnostic and surgical procedures [4]. Animal models of LS reproduce its key pathological characteristics, including hypersensitivity to behavioral effects of VGAs that can be used for a better understanding of risk factors for AiN [7,8,9]. The *ND-23^60114^* and mt:*ND2^del1^* lines of fruit flies (*Drosophila melanogaster*) are models of LS caused by hypomorphic mutations in the nuclearly encoded ND-23 (mammalian Ndufs8) and the mitochondrially encoded mt:ND2 (mammalian ND2) subunits that are conserved proteins of the core of Complex I. They mimic the age-dependent neurodegeneration, mitochondrial abnormalities and shortened lifespan of LS patients [8,10] as well as hypersensitivity to VGAs [4,11].

Young adult *ND-23^60114^* flies present with a phenotype of increased behavioral sensitivity to the VGAs isoflurane (ISO) and sevoflurane (SEVO) replicating mammalian data [11]. At 10–13 days old, *ND-23^60114^* flies develop a lethal phenotype within 24 h after exposure to ISO but not to SEVO [12]. AiN is the likely cause because this phenotype is rescued by neuron-specific overexpression of wild-type ND-23 [12]. Interestingly, heterozygous *ND-23^60114^* flies are asymptomatic and have a normal lifespan indicating haplosufficiency of *ND-23* [8]. However, at 30–35 days of age, *ND-23^60114^*/+ flies develop an AiN-like lethal phenotype after exposure to ISO in 75% O_2_ (hyperoxic ISO), indicating that effects of aging may result in haploinsufficiency of the core Complex I subunit under environmental stress conditions [12].

The present study was designed to test the hypotheses that (i) the VGA halothane (HAL, an alkane as opposed to the ethers ISO and SEVO) causes AiN; and (ii) previously unexamined mutations in nuclearly encoded genes of Complexes I–V are associated with AiN.

Our results indicate that HAL is toxic to the LS-model mutants in subunits ND-23 and mt:ND2. We found that mutation of the accessory subunit ND-SGDH (mammalian NDUFB5) is associated with a lethal phenotype after exposure to hyperoxic ISO. We found that mutations in Complexes II–V did not cause mortality following exposure to hyperoxic ISO. We conclude that mutations in Complex I impart the highest risk for AiN and the risk extends to mutations in accessory (i.e., non-core) subunits. We also discovered that HAL sensitizes wild-type Canton S flies to anoxia. Because anesthetic toxicity is discussed almost exclusively in terms of neurotoxicity, we tested whether a neurotoxic dose of ISO measurably affects the intestine as assessed by intestinal permeability (IP). We found that in contrast to the perfect correlation between traumatic brain injury and IP, mortality due to AiN is not tightly linked to IP.

## 2. Results

### 2.1. Mutants of Complex I Subunits ND-23 and Mt:ND2 Are Sensitive to Halothane Toxicity

We exposed mixed-sex, 11–13-day-old Canton S (wild type), *ND-23^60114^* and mt:*ND2^del1^* flies to 2 h of 1.5% HAL in room air. HAL did not affect mortality in 21% and 75% O^2^ in Canton S flies. In contrast, exposure to HAL increased mortality from a natural attrition rate of 6.8 ± 1.64% and 25.9 ± 3.07% in 21% O_2_ to 21.7 ± 4.97% and 56.6 ± 3.11% in *ND-23^60114^* and mt:*ND2^del1^* flies, respectively (*p* < 0.0001 for both genotypes, unpaired *t*-test, Figure 1). Hyperoxia (75% O^2^) increased mortality from HAL in *ND-23^60114^* flies but the difference to mortality in normoxia was not significant (*p* < 0.16, unpaired *t*-test). However, hyperoxia increased HAL mortality in the mt:*ND2^del1^* line from 56.6 ± 3.11% to 96.4 ± 1.11% (*p* < 0.0001, unpaired *t*-test). These results indicate that HAL is toxic to mitochondrial mutants. Because previous reports showed that ISO-induced AiN in *ND-23^60114^* flies was suppressed by hypoxia, we tested HAL toxicity in 5% O^2^. However, hypoxia did not significantly suppress mortality following HAL exposure in either *ND-23^60114^* or the mt:*ND2^del1^* flies. Finally, mortality resulting from 2 h of anoxia in *ND-23^60114^* and mt:*ND2^del1^* flies was not altered by HAL, but it increased mortality in Canton S flies from 8.1 ± 1.52% to 26.6 ± 6.37% (*p* = 0.018, unpaired *t*-test). We conclude that HAL, while chemically different from ISO, is associated with AiN in mutants of core Complex I subunits. Furthermore, HAL is also toxic under anoxic conditions in wild-type Canton S flies, a phenomenon we have not explored in more depth.

### 2.2. Complex I: A Mutant Allele of ND-SGDH Results in AiN

Complex I (NADH-ubiquinone oxidoreductase) is, with 1 MDa, the largest complex of the mETC consisting of a matrix and a membrane arm. Fourteen core subunits contain the catalytic machinery. These are sufficient for oxidoreductase function and are conserved from bacteria to mammals. The remaining subunits (31–32 in mammals, 28 in *Drosophila*) are accessory and phylum-specific [13]. Complex I is L-shaped with a membrane arm and a matrix arm [14]. About 30% of mitochondrial diseases affecting energy metabolism are caused by mutations in the nuclearly or mitochondrially encoded subunits of Complex I [15]. Some tested mutations sensitize animals to the behavioral effects of VGAs and may represent perioperative risk factors.

*ND-23^60114^* flies showed a strong AiN phenotype at 2 weeks (8–13) days old but not earlier (Figure 2A,A’), while Canton S flies showed no AiN throughout the tested lifespan (Figure 2A).

To investigate the importance of other Complex I subunits for VGA toxicity, we screened the mutants listed in Table 1 (Figure 3).

Mortality of *ND-SGDH^G544^*^7^ mutants was increased by hyperoxic ISO from 11.4 ± 4.82% to 26.8 ± 5.80% (*p* = 0.027, one-way paired *t*-test). For *ND-B22^KG08637^* mortality at 4 weeks also appears increased (but lacked statistical significance, *p*= 0.073, one-way paired *t*-test), which might be due to the low number of biological replicates available at this age. To confirm the positive finding of the screen (Figure 3), we re-tested *ND-SGDH^G544^*^7^ at a single time point: 15–20 days of age (Figure 4). Mortality 24 h after exposure to hyperoxic ISO was increased by 12.8 ± 3.16% (from 38.69 ± 3.16% to 51.49 ± 3.23%; *p* = 0.0254, unpaired *t*-test). We conclude that mutations in accessory subunits of Complex I can increase the risk of AiN. At the same time, the age at which AiN manifests may vary between different alleles of the same gene. For example, the *ND-23^60114^* allele shows early severe AiN while *ND-23^B117^* had increased (but not statistically significantly) mortality at 3 weeks.

### 2.3. The Tested Complex II-V Mutants Did Not Show AiN

To investigate the potential role of mETC Complexes II–V in VGA toxicity, we screened mutants carrying homozygous viable alleles in the Complexes II–V listed in Table 2 (Figure 5). All lines were tested under identical conditions repeatedly (i.e., survivors after each exposure were exposed again at the subsequent time point until their number fell below a preset threshold considered as the minimum for meaningful analysis (see Materials and Methods)). None of the mutants had significantly increased mortality from exposure to hyperoxic ISO (Figure 5A,A’). We conclude that the tested homozygous viable alleles of Complexes II–V do not incur AiN even under hyperoxic conditions and at an advanced age.

### 2.4. AiN Is weakly Associated with Increased Intestinal Permeability

Traumatic brain injury (TBI) causes increased intestinal permeability (IP) in mammals and flies [16,17]. In flies, increased IP can be easily assessed by the ‘smurfing’ assay [18]. In uninjured, aging flies, increased IP is a harbinger of impending death [19]. In the fly TBI model, smurfing is highly predictive of mortality within 24 h after injury [17]. To test the degree to which mortality from AiN is associated with changes in IP, we fed blue-colored food to mixed-sex, wild-type Canton S and to *ND-23^60114^* flies at 10–13 days of age prior to exposing them to ISO. Canton S flies neither smurfed nor died during the observation period (Figure 6). Background mortality in *ND-23^60114^* control group was 6%, smurfing occurred in 0.3%, which is similar to the 0.7% spontaneous smurfing rate reported previously in *w^1118^* flies [17]. After exposure to ISO, 6% of flies smurfed (*p* = 0.0018) and 40.4% died (i.e., exposure to ISO increased IP but the majority of flies died from AiN without an increase in IP) (Figure 6). These data suggest that ISO has minimal effect on IP in mutants susceptible to AiN and most deaths occur without an increase in IP.

## 3. Discussion

The principal findings presented in this manuscript are: (i) the alkane HAL causes AiN in flies harboring mutations of core Complex I subunits; (ii) HAL sensitizes wild-type (Canton S) but not mutant (*ND-23^60114^* and mt:*ND2^del1^*) flies to anoxic injury; (iii) mutations in accessory Complex I subunits are associated with hyperoxic ISO-AiN; (iv) mutations in Complexes II–V are not associated with hyperoxic ISO-AiN; (iii) death from AiN is weakly associated with increased intestinal permeability (IP).

### 3.1. HAL Causes AiN in Mitochondrial Mutants

Most mammalian data on AiN and our previous findings in the fly were obtained using the ether ISO. The current experiments aimed to answer the question of whether AiN in mitochondrial mutants is ISO-specific. Because the research in worms suggests that the behavioral sensitivity of the alkane VGA HAL is affected by mitochondrial mutations similar to that of ISO [2], we used HAL in two mutant fly lines carrying either the mitochondrially encoded mt:*ND2^del1^* allele or the nuclearly encoded *ND-23^60114^* allele. HAL differs from ISO in a number of ways: in contrast to the pungent ether ISO (2-Chloro-2-(difluoromethoxy)-1,1,1-trifluoro-ethane), HAL (2-Bromo-2-chloro-1,1,1-trifluoroethane) is a non-pungent alkane with a lower degree of fluorination and higher lipid solubility. In mammals, HAL undergoes a higher degree of metabolic transformation than ISO [20,21]. Our findings of increased mortality after exposure to HAL in both *ND-23^60114^* and mt:*ND2^del1^* flies indicate that HAL is toxic to flies carrying mutations in the core of Complex I.

Previously, we showed that ISO toxicity in the *ND-23^60114^* model is caused by injury to the brain and is not specific to the *ND-23^60114^* allele [12]. We hypothesize that mortality from HAL is also caused by AiN because of the phenotypic similarity: the dose of HAL was equipotent to that of ISO. Death occurred within 24 h after initial recovery from the anesthetic and mortality was modulated by the O_2_ concentration. Furthermore, neither HAL nor ISO are toxic to Canton S flies under normoxic and hyperoxic conditions. Interestingly, HAL increased mortality in Canton S under hypoxic/anoxic conditions, which is reminiscent of hepatic halothane toxicity in mammals [22,23]. Unexpectedly, HAL did not increase mortality under hypoxic or anoxic conditions in mutant flies, indicating that the significance of HAL’s interaction for toxicity may depend on the integrity of mETC Complex I but in a counterintuitive way, in that the normal function of Complex I predisposes to toxicity while an impaired Complex I does not.

### 3.2. Mutations in Accessory Subunits Sensitize to AiN

In mammals, a mutation in the accessory *NDUFS4* subunit underlies a mouse model of LS and increases the behavioral sensitivity to VGAs [24]. Therefore, we hypothesized that accessory subunits may carry an AiN phenotype. Viable homozygous mutants of *ND-18*, (the fly orthologue of *NDUFS4*) are not available, hence the selection of subunits that were not previously tested for a behavioral anesthetic phenotype [25]. While we did not screen all available alleles of the 46 subunits comprising Complex I, we provide a ‘proof of concept’ by identifying one accessory subunit of Complex I that causes AiN. The *ND-SGDH^G6447^* allele significantly increased mortality at 15–20 days of age (Figure 3 and Figure 4). On the other hand, the *ND-23^B117^* allele showed a weaker (not reaching statistical significance) phenotype at a more advanced age than *ND-23^60114^* (Figure 2 and Figure 3). *ND-23^B117^* is caused by a P-element insertion ~50 bp upstream of the 5′ UTR, probably resulting in a weaker *ND-23* allele than the non-synonymous G199A polymorphism resulting in an amino acid substitution at position 19 that underlies the hypomorphic *ND-23^60114^* phenotype. These experiments suggest that the pharmacodynamic impact of a mutation is determined not only by the subunit and the functional significance of the specific allele: age and environmental conditions can modify the severity and time of onset of the disease phenotype as observed in Leber’s hereditary optic neuropathy [26].

### 3.3. Mutations in Complexes II–V Do Not Result in AiN

Previous work in *C. elegans* indicated that behavioral hypersensitivity to VGAs was not affected by mutations in the mETC unless they affected Complex I and that sensitivity to VGAs correlated with Complex I-dependent oxidative phosphorylation [25]. Our findings using AiN as an endpoint in mutations of subunits not tested by Falk et al. are generally in agreement with their findings: disruption of Complex I function is the determinant for adverse interactions with VGAs. A major limitation of these results is that we tested only single alleles and other alleles may behave differently. However, the tested mutations were functionally significant as illustrated by the high natural attrition and limited lifespan of all tested mutants.

In summary, mutations in various Complex I subunits carry the risk of AiN. Our also indicate that age and environmental conditions influence the phenotypic presentation. Within the stated limitations (see Section 3.5), mutations in Complexes II–V do not appear to be associated with AiN.

### 3.4. AiN Differs from TBI in Its Systemic Impact

It was previously shown that AiN by ISO in *ND-23^60114^* flies was caused primarily by injury to the brain [12]. We examined whether the nature of the toxic insult had commonalities with the systemic pathophysiology associated with TBI, and specifically, with the effect of TBI on IP. At a 24 h mortality level of ±25% after experimental TBI in the *w^1118^* line (a standard laboratory fly strain) [17] that is comparable to mortality from AiN in *ND-23^60114^* flies, smurfing and death correlated tightly. We did not see such a correlation between AiN, IP and death (Figure 6). The increase in IP after exposure to ISO may be due to local effects of ISO on intestinal epithelial tight junctions, analogous to its effect on the blood–brain barrier [27], or to a non-specific toxic effect. However, it is unlikely to be a direct consequence of AiN.

### 3.5. Limitations

We used behaviorally equipotent concentrations of HAL and ISO, which may not be toxicologically equipotent. We also avoided quantitative comparisons of AiN-susceptibility across genotypes because of important confounding parameters: the highly different rates of natural attrition between genotypes, variability in mortality and, specifically for experiments involving HAL (Figure 1), the different effects of O^2^ concentration. We also did not adjust experimental timing to lifespans. As a result, experiments were conducted at a more advanced biological age in mutants than in Canton S, which may affect the penetrance of AiN. Finally, only one allele was tested for components of Complexes II–V, which does not exclude the possibility that other alleles may present a toxicity phenotype under these experimental conditions.

## 4. Materials and Methods

The manuscript adheres to the applicable ARRIVE (Animal Research: Reporting of In Vivo Experiments) reporting guidelines (preclinical animal research). Approval from the Institutional Animal Care and Use Committee was waived.

### 4.1. Fly Lines and Culturing

All flies were cultured at 25 °C on cornmeal molasses food. See Table 1 and Table 2 for information on fly lines used. Fly lines were obtained from the Bloomington Drosophila Stock Center, except for Canton S, which is our laboratory strain; *ND-23^60114^*, which was provided by Barry Ganetzky (UW-Madison) [8] and mt:*ND2,* which was a kind gift from the L. Pallanck lab (University of Washington, Seattle) [10]. Flies were tested as homozygotes.

### 4.2. Anesthesia and O_2_ Exposure

Flies were raised at 25 °C for 2–5 days post-eclosion, transferred to vials of 35 mixed-sex flies using CO_2_ and incubated at 29 °C. Vials were placed on their sides to prevent flies becoming stuck in the food. Flies were transferred to fresh food vials three times per week. On the day of exposure, flies were transferred to 50 mL conical tubes without use of CO_2_ and placed onto the serial anesthesia array (SAA). Each 50 mL conical tube represents a biological replicate [11]. A mixture of N_2_ or O_2_ and air (21% O_2_, 79% N_2_) (Airgas, USA) was used to generate hypoxic (5% O^2^) or hyperoxic (75% O_2_) conditions using a clinical anesthesia machine (Aestiva/5; Datex-Ohemda) and confirmed by an inline internal O_2_ sensor. N_2_ alone was used to generate anoxic (0% O_2_) conditions. Commercial agent-specific vaporizers delivered 1.5% HAL or 2% ISO (Ohmeda Fluotec 4 for HAL and Ohmeda Isotec 5 for ISO). After exposure to the desired conditions for 2 hours, the SAA was flushed with air for 5 minutes. Flies were transferred to fresh food culture vials and incubated at 29 °C for 24 h. Percent mortality was determined by counting the number of dead flies and dividing by the total number of flies per vial. Mortality is a binary readout and was assessed by an unblinded observer. For multiple-exposure experiments, surviving flies were placed on fresh food vials at 29 °C after mortality was scored until the next exposure. Multi-exposed flies were tested at weekly intervals. Each vial of 15–35 flies scored was considered one replicate. When fewer than 15 flies survived in a given vial, the vial was discarded. Experiments on a specific fly line were terminated when fewer than three vials remained. Unexposed flies were handled as above but were not placed on the SAA. Each experiment was conducted on at least three different days.

ΔMortality for single-exposure experiments (Figure 1 and Figure 4) was determined by subtracting the average percent mortality of flies exposed to only O_2_ from the percent mortality of each vial of flies exposed to the same concentration of O_2_ and HAL. ΔMortality for multi-exposure experiments (Figure 2, Figure 3 and Figure 5) was determined by subtracting the percent mortality of each vial of exposed flies from the percent mortality of the unexposed control vial. Significance was determined by paired or unpaired *t*-tests as noted.

### 4.3. Smurfing

Flies were raised at 25 °C for up to three days post eclosion, then transferred to vials of 35 mixed-sex flies using CO_2_ and placed at 29 °C. Vials were placed on their sides to prevent flies becoming stuck in the food. Flies were transferred to fresh food vials three times per week. Flies were aged to 10–13 days old, and then fed a nonabsorbable blue dye (FD&C blue dye no. 1©, Spectrum Chemical MFG Corp, New Brunswick NJ, USA.) at a concentration of 2.5% in standard food for 24 hours prior to exposure. Flies were kept overnight at 29 °C, visually inspected for the presence of blue color in their intestine and excluded from the experiment if no color was seen. They were then exposed to 2% ISO at 21% O_2_ for two hours as described above. Following the exposure, flies were placed back on standard food and incubated at 29 °C. Extravasation of the blue dye into the hemolymph (smurfing) was used as a reporter of IP, as previously described [18,19]. Each vial was scored for percent mortality and percent smurfed flies 24 h after exposure.

## Figures and Tables

**Figure 1 ijms-24-01843-f001:**
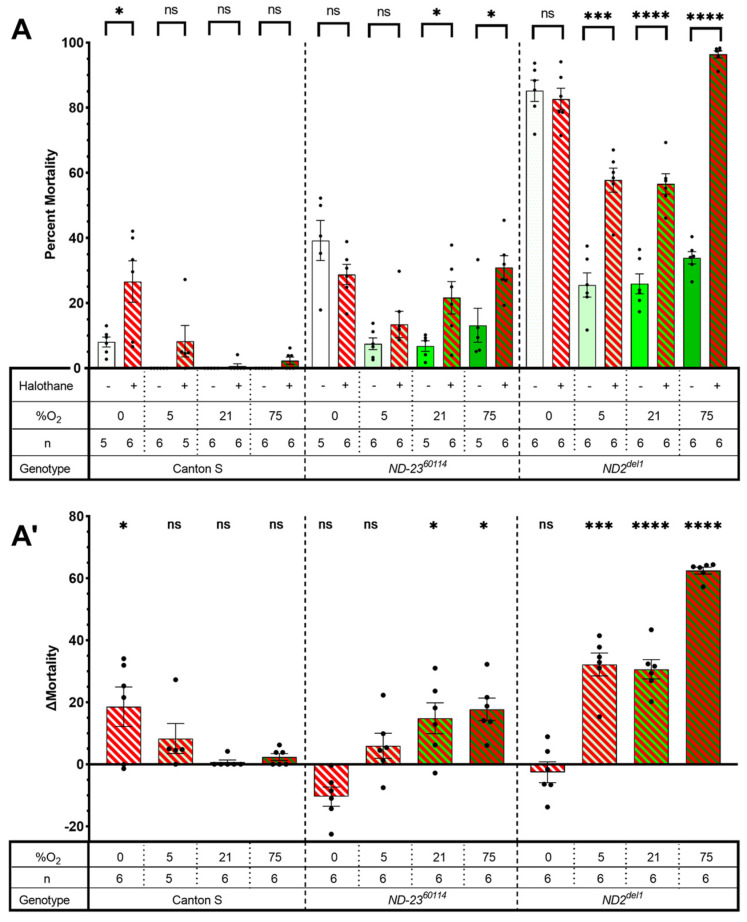
Halothane causes AiN in mitochondrial mutants. Flies of three genotypes at 11 to 13 days of age were exposed to two hours of 1.5% HAL administered by a carrier gas containing between zero and 75% O_2_ and corresponding concentrations of N_2_. The number of dead flies was counted 24 h after exposure. Panel (**A**) shows the percent mortality under the various conditions while panel (**A’**) shows the difference in percent mortality between flies exposed to halothane and those exposed to O_2_ alone. Data shown as mean ±SEM, n indicates number of biological replicates of 25–30 mixed-sex flies contained in a single conical 50 mL tube, each shown as dots. ns: not significant, *: *p* < 0.05, ***: *p* < 0.001, ****: *p* < 0.0001. Note: high rates of natural attrition in mutants.

**Figure 2 ijms-24-01843-f002:**
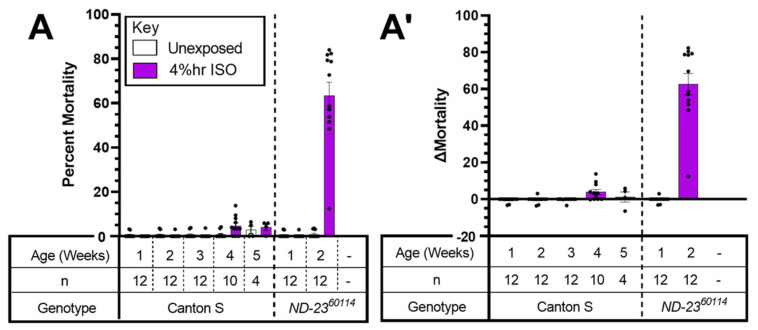
AiN in *ND-23^60114^* flies is age-dependent. Groups of flies were exposed for two hours to 2% ISO in 75% O_2_ at weekly intervals. Mortality was determined 24 h after exposure. No mortality attributable to anesthetic exposure was observed in Canton S flies while *ND-23^60114^* experienced an abrupt increase in mortality at two weeks of age (**A**). (**A’**): same data as A with control mortality subtracted (Δ Mortality). Each dot represents a biological replicate of 15–35 mixed-sex flies. Results are presented as mean ± SEM. Dashes indicate that the experiment was not performed because there were fewer than three vials of 15 flies.

**Figure 3 ijms-24-01843-f003:**
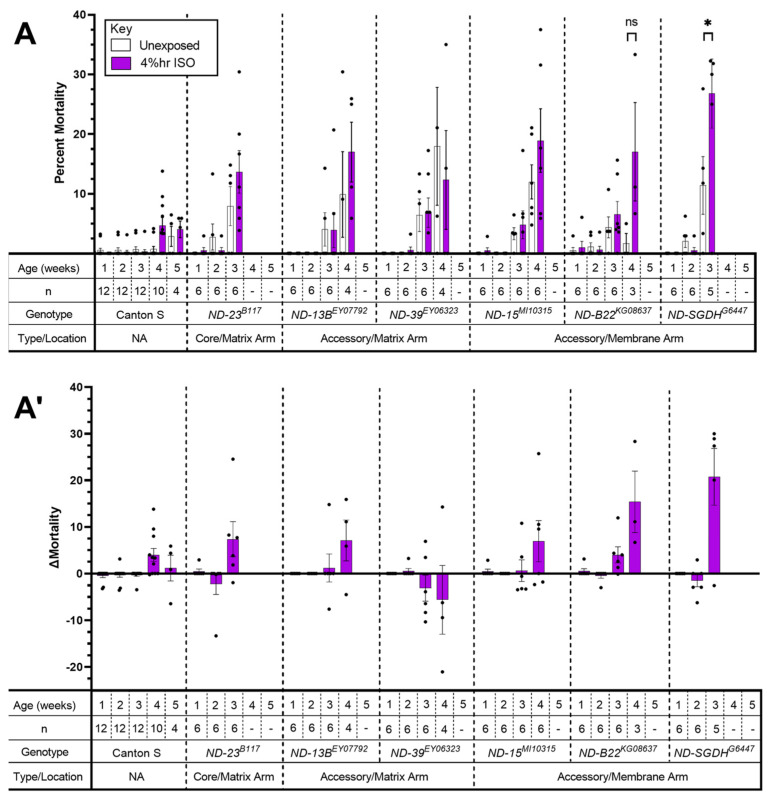
Complex I subunits screen identifies an *ND-SGDH* mutant as AiN susceptible. (**A**). Flies were exposed for two hours to 2% ISO in 75% O_2_ at weekly intervals beginning at one week of age. After exposure, they were returned to standard culture vials. Dead flies were counted and survivors were transferred to fresh vials 24 h after exposure. Mutations in Complex I subunits carried variable age-dependent mortality due to natural attrition (white bars). The increase in mortality after hyperoxic ISO reached significance only for *ND-SGDH^G6447^* at three weeks old. (**A’**). Data from A, control mortality subtracted (Δ Mortality). (Week 1 = 1–6 days old; Week 2 = 8–13 days old; Week 3 = 15–20 days old; Week 4 = 22–27 days old; Week 5 = 29–34 days old.) Each dot represents a biological replicate of 15–30 mixed-sex flies. Results are presented as mean ± SEM. ns: not significant, *: *p* < 0.05 Dashes indicate that the experiment was not performed because there were fewer than three vials of 15 flies.

**Figure 4 ijms-24-01843-f004:**
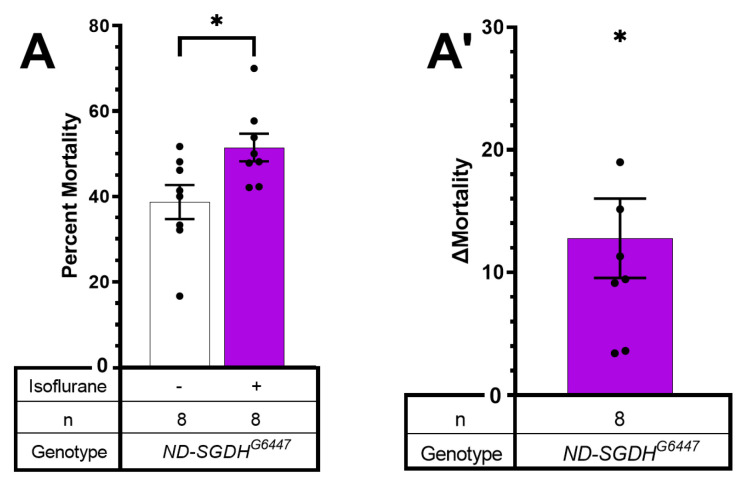
Hyperoxic ISO increases 24 h mortality in *ND-SGDH^G6447^* mutants. (**A**). Flies were exposed to 2% ISO in 75% O_2_ for two hours at 15–20 days old, returned to culture vials and scored 24 h after exposure. Mortality increased in mutants but not in wild-type flies (*p* = 0.0254, unpaired *t*-test). (**A’**). Data as in A, control mortality subtracted (Δ Mortality). Each dot represents a biological replicate of 15–35 mixed-sex flies. Results presented as mean ± SEM. * *p* < 0.05.

**Figure 5 ijms-24-01843-f005:**
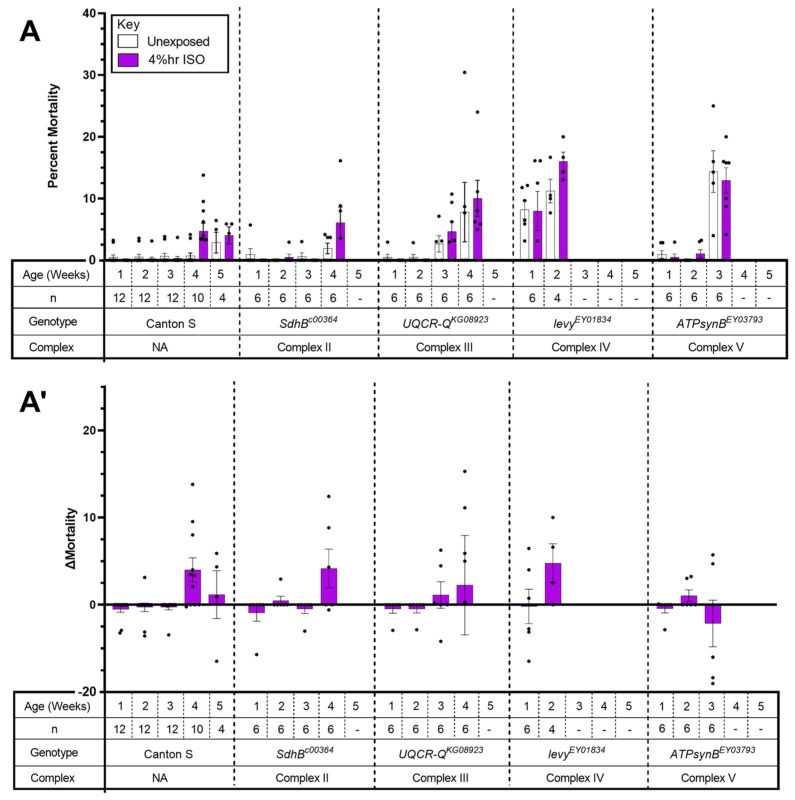
Hyperoxic ISO does not increase 24 h mortality in mutants of Complexes II–V. (**A**). Repeated exposure at weekly intervals for two hours to 2% ISO in 75% O_2_ at 1–5 weeks of age. Dead flies were counted and survivors were transferred to fresh vials 24 h after exposure. Results presented as mean ± SEM. (**A’**). Data as in A with control mortality subtracted (Δ Mortality). Each dot represents a biological replicate of 15–35 mixed-sex flies. Dashes indicate that the experiment was not performed because there were fewer than three vials of 15 flies. (Week 1 = 1–6 days old; Week 2 = 8–13 days old; Week 3 = 15–20 days old; Week 4 = 22–27 days old; Week 5 = 29–34 days old.)

**Figure 6 ijms-24-01843-f006:**
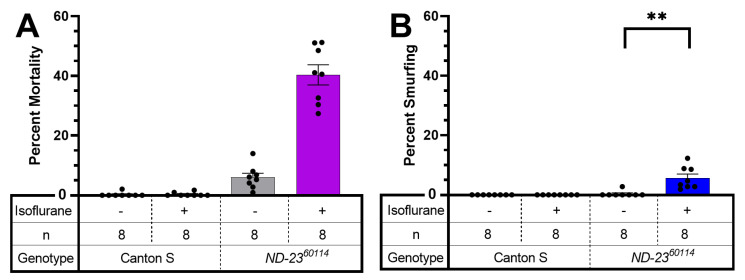
Mortality from AiN is not a result of increased intestinal permeability (IP). (**A**). Normoxic ISO increases mortality 24 h after exposure in *ND-23^60114^* but not in wild-type flies. Note: unexposed *ND-23^60114^* incur a background mortality over the observation period. (**B**). No Canton S flies showed increased IP after exposure to ISO, but a small fraction of *ND-23^60114^* flies (close to the background mortality without ISO exposure) smurfed. Flies of 1-10 days of age were fed a nonabsorbable blue dye for 24 h prior to exposure. Dead and smurfed flies were counted 24 h after exposure. Each dot represents a biological replicate of 15–35 mixed-sex flies. Results presented as mean ± SEM. **: *p* < 0.01.

**Table 1 ijms-24-01843-t001:** Complex I alleles screened for hyperoxic ISO toxicity. BDSC: Bloomington Drosophila Stock Center.

Drosophila Gene and Allele	Mammalian Name	BDSC Stock Number	Insertion Location/Description of Mutation	Type/Location of Subunit
mt:*ND2^del1^*	*ND2*	NA	9 bp deletion destroys the BglII restriction enzyme site	Core/Membrane Arm
*ND-23^60114^*	*NDUFS8*	NA	G199A Amino Acid Substitution	Core/Matrix Arm
*ND-23^B117^*	*NDUFS8*	16143	~50 bp upstream of 5′ UTR	Core/Matrix Arm
*ND-13B^EY07792^*	*NDUFA5*	16860	5′ UTR	Accessory/Matrix Arm
*ND-39^EY06323^*	*NDUFA9*	15821	Intron/~50 bp upstream of 5′ UTR	Accessory/Matrix Arm
*ND-15^MI10315^*	*NDUFS5*	53837	5′ UTR/Intron	Accessory/Membrane Arm
*ND-B22^KG08637^*	*NDUFB9*	15134	5′ UTR	Accessory/Membrane Arm
*ND-SGDH^G6447^*	*NDUFB5*	27208	5′ UTR	Accessory/Membrane Arm

**Table 2 ijms-24-01843-t002:** Alleles of Complexes II–V screened for hyperoxic ISO toxicity. BDSC: Bloomington Drosophila Stock Center.

Drosophila Gene	Mammalian Name	BDSC Stock Number	Insertion Location/Description of Mutation	Complex
*SdhB^c00364^*	*SdhB*	10039	5′ UTR	Complex II
*UQCR-Q^KG08923^*	*UQCRQ*	16470	Intron	Complex III
*Levy^EY01834^*	*COX6A*	15080	5′ UTR	Complex IV
*ATPsynB^EY03793^*	*ATP5PB*	16575	5′ UTR	Complex V

## Data Availability

The authors confirm that data supporting the findings of this study are available within the article. Raw data supporting the findings of this study are available from the corresponding author on request.

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
