# Peer review of "Mutations in Complex I of the Mitochondrial Electron-Transport Chain Sensitize the Fruit Fly (Drosophila melanogaster) to Ether and Non-Ether Volatile Anesthetics"

_ijms, 2023, doi:10.3390/ijms24031843_

Round 1

Reviewer 1 Report

In this study the authors investigated in the fruit fly the effect of mutations to the mitochondrial electron chain on anaesthetic sensitivity and neurotoxicity, focusing on the non-ether halothane and accessory subunits of Complex I. The findings extend recent investigations in this area by showing that anaesthetic-induced neurotoxicity is Complex I dependent (core and accessory mutations), but not limited to ether anaesthetics. Overall, I found the study interesting (once I got to grips with the methods) and relevant to the ongoing discussion around anaesthetic-related disruption of neurometabolism.

My main criticism of the paper is that the methodology is difficult to follow with the current format. The methodology section is extremely brief and does not adequately explain the details of the experiments. Instead, there are snippets of methodological details dotted throughout the results, including some figure legends. This is OK, but these details also need to be consolidated into the methods section, so that the reader can get a complete overview of the experimental details without having to pick them out of the results. This is particularly important for the reader who is not familiar with these types of experiments.

Most of the rest of my comments relate to relatively minor issues around clarification of details.

Readers not familiar with fruit fly research will not understand that the Canton strain is the wildtype control. It would be helpful to make this clearer (and probably earlier than the first mention on ln 73).

I would remove the reference to the IP results at the end of the Introduction, because there is no context for this here. The result is interesting but can be adequately explained in the body of the results section.

Please clarify that “biological replicates” refers to the 50mL conical tubes introduced in the methods.

Figure 1. Please add significance bars – or some indication of significant differences between groups.

What is the concentration of CO2 (ln 294)

Why was 75% O2 chosen for the hyperoxic group? This is higher than would normally be given clinically for a volatile anaesthetic. Clearly oxygen is a confounder, so this level is important.

All of ln 101-109 is out of place in the results.

Ln 101-102. This sentence does not make sense.

Ln 104. These numbers do not add up. 14 core subunits + 28 remaining subunits = 42 subunits total (not 44 or 45). Please clarify these numbers.

Ln 110. Bracket in the wrong place.

Section 2.2. Standardise terminology. Is “peripheral arm” (ln 102) equivalent to “matrix arm” (ln 106)?

Ln 107. Nuclearly??

Figure 2. It’s not clear which are anaesthetic exposed and which are not. Presumably the figure follows the same layout as Figure 1, but this is not clear. Also, if there is “no mortility” in the control group, why do the number of vials drop from 12 to 4 from 3-5 weeks. Presumably you mean there is no increase in mortality due to anaesthetic exposure. This needs to be made clearer.

Why was 4 vials considered the threshold for meaningful analysis?

Figure 3. Do the dashes (“-“) represent zero remaining vials?

Reviewer 2 Report

The authors have conducted a follow-up study using Drosophila as a model organism where they investigate the role of mitochondrial electron transport chain subunits in anaesthetic induced neurotoxicity. The authors have previously demonstrated that isoflurane exposure causes lethality through neuronal mechanisms using an already characterised Drosophila mutant of the mitochondrial Complex I. Here, they contribute additional evidence that similar to ether anaesthetic, exposure to non-ether anaesthetic halothane can result in premature lethality in the ND-23[60114] Complex I mutant in oxygen dependent manner. Authors then used existing less or not at all characterised alleles in Drosophila that localize in non-coding sequences of genes encoding components of either Complex I or other complexes of the electron transport chain. Here, the authors conclude that mutations in Complex I however not those in Complex II-IV can cause anaesthetic induced toxicity. This observation, if true, would resemble previous findings in C. elegans and indicate that this mechanism is conserved. Further the authors, investigate to whether the anaesthetic induced toxicity and death of the organism is associated with increased intestinal permeability similar to that observed in traumatic brain injury, which is not the case.

In this manuscript authors contribute some additional important observations related to conserved mechanisms of anaesthetic induced toxicity. This work uses Drosophila as a powerful genetic model organism, which expands the already existing foundation and the innovative methodology used by this lab will be of great importance. However, there are several points that would need to be addressed in order to consolidate some of the conclusions.

Major points –

a)       The halothane induced toxicity in Canton S in anoxic conditions as presented in figure 1, is then discussed as resembling liver toxicity in patients on page 9 LL224-226. Could the authors please provide more logical arguments to corroborate this correlation. And if that was the case how do they relate this to anaesthetic induced neurotoxicity hypothesis. Can the lethality within 24 hrs support hepatotoxic-like neurotoxicity? How would  the CNS specific rescue of isoflurane dependent phenotype in ND-23[60114] from their previous publication fit into the hypothesis.

b)      Is the halothane induced lethality in fly ND-23[60114] mutant due to neurotoxicity?

c)       The lack of phenotype in experiments presented in Figures 3 and 5 may be explained by isomorphic nature of the less characterised insertion alleles. In order to support their conclusions (including the title) authors would need to consolidate that the observations in Figure 5 in particular are with certainty not false negative, and to provide more genetic characterisation of those mutants. The number of alleles studied in Figure 5 is very low – authors cannot exclude that they may have been “unlucky”, in contrast to their “lucky” albeit weak find in figure 3. Authors could for instance assess the levels of gene expression (at least at mRNA level if antibodies aren’t available, which may be the case for some of genes) or at least demonstrate a presence of a respiratory chain phenotype to confirm hypo-, hyper or amorphic nature of allele. This will be also important in order to understand the absence of the phenotype in the ND-23[B117] strain, which the authors argue is due to the insertion in the 5’UTR. Indeed, most of the alleles studied here are in the 5’UTR. Perhaps authors could also consider to take advantage of the binary expression system in the fly to silence or over express some of the genes in combination with siRNA or the here studied EY library alleles. This may also benefit the consolidation of neurological defects specifically.

d)      I agree with authors argument to the limitation of the study and the natural lethality of the flies.  P 5 L 144 – similar repetitions may be performed with ND-23B117, at least

e)      The importance of the observation that there is no intestinal leakage may be discussed in more detail.

Minor points:

A)      Throughout the manuscript: the citations typically are placed before the punctuation [1].

B)      Page 2 L 47 – specify behavioural phenotypes at least once in the manuscript

C)      Intestinal permeability and relevance to AIN should be introduced in the Introduction

D)      P2  L 87 – citation missing

E)       P2- LL 88-90 – the authors could highlight that halothane did not exert additional toxicity in ND-23[60114] mutant. What would be the explanation?

F)       P3 L98 – groups of flies rather than biological replicates

G)      P3 LL 101-106 – can be formulated more concise and clear.

H)      P3 LL110 – 112 – falls out of context here.

I)        P4 L125 – delt morality doesn’t add anything here. However in figure 1 it could actually highlight the absence of effect of Halothane on mutants at 0% oxygen.

J)        Figure 3 – It may be helpful for easier understanding if the mutants were organised based on subunit organisation (which could be included in the description – similar as mutants for different complexes were organised in Figure 5).

K)      P5 L 161 – The hemizygous allele should be studied without balancer or a balancer only control should be provided.

L)       P 6 L150 – statistically non-significant observations are considered as potentially serendipitous and therefore should be concluded as unchanged
